# Identifying Products in Online Cybercrime Marketplaces:
# A Dataset and Fine-grained Domain Adaptation Task

## Abstract

One weakness of machine-learned NLP models is that they typically perform poorly on out-of-domain data. In this work, we study the task of identifying products being bought and sold in online cybercrime forums, which exhibits particularly challenging cross-domain effects. We formulate a task that represents a hybrid of slot-filling information extraction and named entity recognition and annotate datasets consisting of data from four different forums. Each of these forums constitutes its own "fine-grained domain" in that the forums cover different market sectors with different properties, even though all forums are in the broad domain of cybercrime. We characterize these domain differences in the context of a learning-based system: supervised models see decreased accuracy when applied to new forums, and standard techniques for semi-supervised learning and domain adaptation have limited effectiveness on this data, which suggests the need to improve these techniques.

We release a dataset of 93,924 posts from across 4 forums, with annotations for 1,938 posts.[1]

## 1 Introduction

NLP can be extremely useful for enabling scientific inquiry, helping us to quickly and efficiently understand large corpora, gather evidence, and test hypotheses (Bamman et al., 2013; O'Connor et al., 2013). One domain for which automated analysis is particularly useful is Internet security: researchers obtain large amounts of text data pertinent to active threats or ongoing cybercriminal activity, for which the ability to rapidly characterize that text and draw conclusions can reap major benefits (Krebs, 2013a,b). However, conducting automatic analysis is difficult because this data is out-of-domain for conventional NLP models, which harms the performance of both discrete models (McClosky et al., 2010) and deep models (Zhang et al., 2017). Not only that, we show that data from one cybercrime forum is even out of domain with respect to *another* cybercrime forum, making this data especially challenging.

In this work, we present the task of identifying products being bought and sold in the marketplace sections of these online cybercrime forums. We define a token-level annotation task where, for each post, we annotate references to the product or products being bought or sold in that post. Having the ability to automatically tag posts in this way lets us characterize the composition of a forum in terms of what products it deals with, identify trends over time, associate users with particular activity profiles, and connect to price information to better understand the marketplace. Some of these analyses only require post-level information (what is the product being bought or sold in this post?) whereas other analyses might require token-level references; we annotate at the token level to make our annotation as general as possible. Our dataset has already proven enabling for case studies on these particular forums (anon. work in press).

Our task has similarities to both slot-filling information extraction (with provenance information) as well as standard named-entity recognition (NER). Compared to NER, our task features a higher dependence on context: we only care about the specific product being bought or sold in a post, not other products that might be mentioned. Moreover, because we are operating over forums, the data is substantially messier than classical NER corpora like CoNLL (Tjong Kim Sang and De Meulder, 2003). While prior work has dealt with these messy characteristics for syntax (Kaljahi et al., 2015) and for discourse (Lui and Baldwin, 2010; Kim et al., 2010; Wang et al., 2011), our work is the first to tackle forum data (and marketplace forums specifically) from an information extraction perspective.

Having annotated a dataset, we examine basic supervised learning approaches to the product extraction problem. Simple binary classification of tokens as product names can be effective, but performance drops off precipitously when a system trained on one forum is applied to a different fo-

---

[1] Available upon publication, along with all models and code from this work.

| Forum | Posts | Words per post | Products per post | Annotated posts | Annotators per post | Inter-annotator agreement 3-annotated | all-annotated |
|---|---|---|---|---|---|---|---|
| Darkode | 3,368 | 61.5 | 3.2 | 660/100/100 | 3/8/8 | 0.62 | 0.66 |
| Hack Forums | 51,271 | 58.9 | 2.2 | 758/140 | 3/4 | 0.58 | 0.65 |
| Blackhat | 167 | 174 | 3.2 | 80 | 3 | 0.66 | 0.67 |
| Nulled | 39,118 | 157 | 2.3 | 100 | 3 | 0.77 | - |

Table 1: Forum statistics. The left columns (posts and words per post) are calculated over all data, while the right columns are based on annotated data only. Slashes indicate the train/development/test split for Darkode and train/test split for Hack Forums. Agreement is measured using Fleiss' Kappa; the two columns cover data where three annotators labeled each post and a subset labeled by all annotators.

rum: in this sense, even two different cybercrime forums seem to represent different "fine-grained domains." Since we want to avoid having to annotate data for every new forum that might need to be analyzed, we explore several methods for adaptation, mixing type-level annotation (Garrette and Baldridge, 2013; Garrette et al., 2013), token-level annotation (Daume III, 2007), and semi-supervised approaches (Turian et al., 2010).[2] We find little improvement from these methods and discuss why they fail to have a larger impact.

Overall, our results characterize the challenges of our fine-grained domain adaptation problem in online marketplace data. We believe that this new dataset provides a useful testbed for additional inquiry and investigation into modeling of fine-grained domain differences.

## 2 Dataset and Annotation

We consider several forums that vary in the nature of products being traded:

- Darkode: Cybercriminal wares, including exploit kits, spam services, ransomware programs, and stealthy botnets.

- Hack Forums: A mixture of cyber-security and computer gaming blackhat and non-cybercrime products.

- Blackhat: Blackhat Seach Engine Optimization techniques.

- Nulled: Data stealing tools and services.

Table 1 gives some statistics of these forums.

Figure 1 shows two examples of posts from Darkode. In addition to aspects of the annotation, which we will discuss shortly, we see that

---

[2]Of course, these techniques can also be combined in a single system (Kshirsagar et al., 2015).

0-initiator4856

TITLE: [ buy ] Backconnect bot
BODY: Looking for a solid backconnect bot .
    If you know of anyone who codes them please let me know

0-initiator6830

TITLE: Coder
BODY: Need sombody too mod DCIBot for me add the
    following :
    Update Cmd
    Autorun Obfuscator ( each autorun diffrent & fud )
    Startup Mod ( needs too work on W7/VISTA )
    Pm .

Figure 1: Example post and annotations from Darkode, with one sentence per line. Annotated product tokens are underlined. The first example is quite short and straightforward. The second exhibits our annotations of both the core product (*mod DCIBot*) and the method for obtaining that product (*sombody*).

the text exhibits common features of web text: abbreviations, ungrammaticality, spelling errors, and visual formatting, particularly in thread titles. Additionally, note how some words are present that might be products in other contexts, but are not here (e.g., *obfuscator*).

### 2.1 Annotation Process

We developed our annotation guidelines through six preliminary rounds of annotation, covering 560 posts. Each round was followed by discussion and resolution of every post with disagreements. We benefited from members of our team who brought extensive domain expertise to the task. As well as refining the annotation guidelines, the development process trained annotators who were not security experts. The data annotated during this process is not included in Table 1.

Once we had defined the annotation standard,

we annotated datasets from Darkode, Hack Forums, Blackhat, and Nulled as described in Table 1.[3] Every post in the Darkode training, Hack Forums training, Blackhat test, and Nulled test sets was annotated by three people; these annotations were then merged into a final annotation based on majority vote. The development and test sets for Darkode and Hack Forums were annotated by addition team members (five for Darkode, one for Hack Forums), and then every disagreement was discussed and resolved to produce a final annotation. The annotation was performed entirely by the authors, who are researchers in either NLP or computer security.

We preprocessed the data using the tokenizer and sentence-splitter from the Stanford CoreNLP toolkit (Manning et al., 2014). Note that many sentences in the data are already delimited by line breaks, making the sentence-splitting task slightly easier. We performed annotation on the tokenized data so that annotations would be consistent with surrounding punctuation and hyphenated words.

Our full annotation guide will be available in supplementary material. Our basic annotation principle is to annotate tokens when they are either the product that will be delivered or are an integral part of the method leading to the delivery of that product. For example, in Figure 1, the fragment *mod DCIBot* is the core product, but *coder* is annotated as well, since this person is inextricably linked to the service being provided: the post seeks someone to perform the task. However, the *Backconnect bot* example is a more straightforward transaction (the human agent is less integral), so only the product is annotated. Verbs may also be annotated, as in the case of *hack an account*: here *hack* is the method and the deliverable is the *account*, so both are annotated.

When the product is a multiword expression (e.g., *Backconnect bot*), it is almost exclusively a noun phrase, in which case we annotate the head word of the noun phrase (*bot*). Annotating single tokens instead of spans meant that we avoided having to agree on an exact parse of each post, since even the boundaries of base noun phrases can be quite difficult to agree on in ungrammatical text.

If multiple different products are being bought/sold, we annotate them all. We do not annotate:

- Features of products, e.g., *Update Cmd* in Figure 1.
- Generic product references, e.g., *this*, *them*.
- Product mentions inside "vouches" (reviews from other users).
- Product mentions outside of the first and last 10 lines of each post.[4]

Table 1 shows interannotator agreement according to our annotation scheme. We use the Fleiss' Kappa measurement (Fleiss, 1971), treating our task as a token-level annotation where every token is annotated as either a product or not. We chose this measure as we are interested in agreement between more than two annotators (ruling out Cohen's kappa), have a binary assignment (ruling out correlation coefficients) and have datasets large enough that the biases Krippendorff's Alpha addresses are not a concern. The values indicate reasonable agreement.

## 2.2 Discussion

Because we annotate entities in a context-sensitive way (i.e., only annotating those in product context), our task resembles a post-level information extraction task. The product information in a post can be thought of as a list-valued slot to be filled in the style of TAC KBP (Surdeanu, 2013; Surdeanu and Ji, 2014), with the token-level annotations constituting provenance information. However, we chose to anchor the task fully at the token level to simplify the annotation task: at the post level, we would have to decide whether two distinct product mentions were actually distinct products or not, which requires heavier domain knowledge. Our approach also resembles the fully token-level annotations of entity and event information in the ACE dataset (NIST, 2005).

## 3 Evaluation Metrics

In light of the various views on this task and its different requirements for different potential applications, we describe and motivate a few distinct evaluation metrics below. The choice of metric will impact system design, as we discuss in the following sections.

---

[3] Not pictured is the fact that we annotated small numbers of Darkode training, Hack Forums training, and Blackhat test posts with all annotators in order to simply check agreement.

[4] In preliminary annotation we found that content in the middle of the post typically described features or gave instructions, without explicitly mentioning the product. Most posts are unaffected by this rule (96% of Darkode, 77% of Hack Forums, 84% of Blackhat, and 93% of Nulled), but it still substantially reduced annotator effort because the posts it does affect are quite long – 61 lines on average.

**Token-level accuracy**  We can follow the approach used in token-level tasks like NER and compute precision, recall, and $F_1$ over the set of tokens labeled as products. This most closely mimics our annotation process.

**Type-level product extraction (per post)**  For many applications, the primary goal of the extraction task is more in line with KBP-style slot filling, where we care about the set of products extracted from a particular post. Without a domain-specific lexicon containing full synsets of products we care about (e.g., something that could recognize the synonymity of *hack* and *access*), it is difficult to evaluate this in a fully satisfying way. However, we can approximate this evaluation by comparing the set of (lowercased, stemmed) product *types* in a post with the set of product types predicted by the system. Again, we can consider precision, recall, and $F_1$ over these two sets. This metric favors systems that consistently make correct post-level predictions even if they do not retrieve every token-level occurrence of the product.

**Post-level accuracy**  Our type-level extraction will naturally be a conservative estimate of performance simply because there may seem to be multiple "products" that are actually just different ways of referring to one core product. Roughly 60% of posts in the two forums contain multiple annotated tokens that are distinct beyond stemming and lowercasing. However, we analyzed 100 of these multiple product posts across Darkode and Hack Forums, and found that only 6 of them were actually selling multiple products, indicating that posts selling multiple types of products are actually quite rare (roughly 3% of cases overall). In the rest of the cases, the variations were due to slightly different ways of describing the same product.

In light of this, we also might consider asking the system to extract *some* product reference from the post, rather than all of them. Specifically, we compute accuracy on a post-level by checking whether a single product type extracted by the system is contained in the annotated set of product types. Because most posts feature one product, this metric is sufficient to evaluate whether we understood what the core product of the post was.

### 3.1 Phrase-level Evaluation

Another axis of variation in metrics comes from whether we consider token-level or phrase-level outputs. As noted in the previous section, we did not annotate noun phrases, but we may actually be interested in identifying them. In Figure 1, for example, extracting *Backconnect bot* is more useful than extracting *bot* in isolation, since *bot* is a less specific characterization of the product.

We can convert our token-level annotations to phrase-level annotations by projecting our annotations to the noun phrase level based on the output of an automatic parser. We used the parser of Chen and Manning (2014) to parse all sentences of each post. For each annotated token that was given a nominal tag (N*), we projected that token to the largest NP containing it of length less than or equal to 7; most product NPs are shorter than this, and when the parser predicts a longer NP, our analysis found that it typically reflects a mistake. In Figure 1, the entire noun phrase *Backconnect bot* would be labeled as a product. For products realized as verbs (e.g., *hack*), we leave the annotation as the single token.

Throughout the rest of this work, we will evaluate sometimes at the token-level and sometimes at the NP-level[5] (including for the product type evaluation and post-level accuracy); we will specify which evaluation is used where.

## 4 Models

We consider several simple baselines for product extraction as well as two learning-based methods.

**Baselines**  A simple **frequency** baseline is to take the most frequent noun or verb in a post and classify all occurrences of that word type as products. A more sophisticated lexical baseline is based on a product **dictionary** extracted from our training data: we tag the most frequent noun or verb in a post that also appears in this dictionary. This method fails primarily in that it prefers to extract common words like *account* and *website* even when they do not occur as products. Finally, we can tag the **first** noun phrase of the post as a product, which will often capture the product if it is mentioned in the title of the post.[6]

**Binary classifier**  One learning-based approach to this task is to employ a binary SVM classifier

---

[5]Where NP-level means "noun phrases and verbs" as described in Section 3.1.

[6]Since this baseline fundamentally relies on noun phrases, we only evaluate it in the noun phrase setting.

for each token in isolation.[7] Our features look at both the token under consideration as well as neighboring tokens, as described in the next paragraph. A vector of "base features" is extracted for each of these target tokens: these include 1) sentence position in the document and word position in the current sentence as bucketed indices; 2) word identity (for common words), POS tag, and dependency relation to parent for each word in a window of size 3 surrounding the current word; 3) character 3-grams of the current word. The same base feature set is used for every token.

Our token-classifying SVM extracts base features on the token under consideration as well as its syntactic parent. Before inclusion in the final classifier, these features are conjoined with an indicator of their source (i.e., the current token or the parent token). Our NP-classifying SVM extracts base features on first, last, head, and syntactic parent tokens of the noun phrase, again with each feature conjoined with its token source.

We weight false positives and false negatives differently to adjust the precision/recall curve (tuned on development data for each forum), and we also empirically found better performance by upweighting the contribution to the objective of singleton products (product types that occur only once in the training set).

**Post-level classifier** As discussed in Section 3, one metric we are interested in is whether we can find *any* occurrence of a product in a post. This task may be easier than the general tagging problem: for example, if we can effectively identify the product in the title of a post, then we do not need to identify additional references to that product in the body of the post. Therefore, we also consider a post-level model, which directly tries to select one token (or NP) out of a post as the most likely product. Structuring the prediction problem in this way naturally lets the model be more conservative in its extractions and simplifies the task, since highly ambiguous product mentions can be ignored if a clear product mention is present. Put another way, it supplies a form of prior knowledge that can be useful for the task, namely that each post has exactly one product.

Our post-level system is formulated as an instance of a latent SVM (Yu and Joachims, 2009). The output space is the set of all tokens (or noun

---

[7]This performs similarly to using a token-level CRF with a binary tagset.

| | Token Prediction | | | | | | |
|---|---|---|---|---|---|---|---|
| | Tokens | | | Products | | | Posts |
| | P | R | $F_1$ | P | R | $F_1$ | Acc. |
| Freq | 41.9 | 42.5 | 42.2 | 48.4 | 33.5 | 39.6 | 45.3 |
| Dict | 57.9 | 51.1 | 54.3 | 65.6 | 44.0 | 52.7 | 60.8 |
| Binary | 62.4 | 76.0 | **68.5** | 58.1 | 77.6 | 66.4 | 75.2 |
| Post | 82.4 | 36.1 | 50.3 | 83.5 | 56.6 | **67.5** | **82.4** |
| | NP Prediction | | | | | | |
| | NPs | | | Products | | | Posts |
| | P | R | $F_1$ | P | R | $F_1$ | Acc. |
| Freq | 61.8 | 27.9 | 38.4 | 61.8 | 50.0 | 55.2 | 61.8 |
| Dict | 59.1 | 61.3 | 60.2 | 70.1 | 56.6 | 62.6 | 67.0 |
| First | 73.2 | 33.0 | 45.5 | 73.2 | 59.2 | 65.4 | 73.2 |
| Binary | 65.6 | 79.0 | **71.7** | 61.7 | 87.5 | 72.4 | 85.5 |
| Post | 89.6 | 40.4 | 55.7 | 89.6 | 72.5 | **80.1** | **89.6** |

Table 2: Development set results on Darkode. Bolded values represent statistically-significant improvements over all other values in the column with $p < 0.05$ according to a bootstrap resampling test. Our post-level system outperforms our binary classifier at whole-post accuracy and on type-level product extraction, even though it is less good on the token-level metric. All systems are consistently better at identifying product NPs than at identifying product tokens.

phrases, in the NP case) in the post. The latent variable is the choice of token/NP to select, since there may be multiple annotated tokens. The features used on each token/NP are the same as in the token classifier.

We trained all of the learned models by subgradient descent on the primal form of the objective (Ratliff et al., 2007; Kummerfeld et al., 2015). We use AdaGrad (Duchi et al., 2011) to speed convergence in the presence of a large weight vector with heterogeneous feature types. All product extractors in this section are trained for 5 iterations with $\ell_1$-regularization tuned on the development set.

## 4.1 Basic Results

Table 2 shows development set results on Darkode for each of the four systems for each metric described in Section 3. Our learning-based systems substantially outperform the baselines on the metrics they are optimized for. The post-level system underperforms the binary classifier on the token evaluation, but is superior at not only post-level accuracy but also product type $F_1$. This lends credence to our hypothesis that picking one product suffices to characterize a large fraction of posts.

When evaluated on noun phrases, we see that the systems generally perform better, an unsurprising result given that this evaluation is more

| | Darkode | | | Hack Forums | | | Blackhat | | | Nulled | | | Avg |
| System | P | R | $F_1$ | P | R | $F_1$ | P | R | $F_1$ | P | R | $F_1$ | $F_1$ |
| | Trained on Darkode | | | | | | | | | | | | |
| Dictionary | 57.3 | 54.3 | 55.8 | 44.8 | 40.8 | 42.7 | 42.5 | 39.0 | 40.7 | 57.3 | 43.9 | 49.7 | 47.2 |
| Binary | 75.4 | 79.8 | 77.6 | 49.2 | 49.7 | 49.5 | 57.5 | 59.1 | 58.3 | 56.9 | 64.1 | 60.3 | 61.4 |
| Binary + Brown Clusters | 76.8 | 79.4 | 78.1 | 52.1 | 49.1 | 50.6 | 58.5 | 60.9 | 59.7 | 60.5 | 63.2 | 61.8 | 62.6 |
| Binary + Gazetteers | 73.8 | 77.7 | 75.7 | 52.3 | 51.9 | 52.1 | – | – | – | – | – | – | – |
| | Trained on Hack Forums | | | | | | | | | | | | |
| Dictionary | 57.7 | 44.4 | 50.2 | 51.4 | 51.9 | 51.7 | 45.6 | 44.8 | 45.2 | 53.8 | 43.4 | 48.1 | 48.8 |
| Binary | 68.9 | 58.4 | 63.2 | 58.3 | 62.6 | 60.4 | 60.8 | 56.3 | 58.5 | 76.4 | 66.8 | 71.2 | 63.3 |
| Binary + Brown Clusters | 69.5 | 57.2 | 62.7 | 59.2 | 64.1 | 61.6 | 60.9 | 57.4 | 59.1 | 78.1 | 65.9 | 71.5 | 63.7 |
| Binary + Gazetters | 65.1 | 64.6 | 64.8 | 59.5 | 60.4 | 60.0 | – | – | – | – | – | – | – |

Table 3: Test set results at the NP level in within-forum and cross-forum settings for a variety of different systems. Using either Brown clusters or gazetteers gives mixed results on cross-forum performance: none of the improvements are statistically significant with $p < 0.05$ according to a bootstrap resampling test. Gazetteers are unavailable for Blackhat and Nulled since we have no training data for those forums.

forgiving (token distinctions within noun phrases are erased). The post-level NP system achieves an F-score of roughly 80 on product type identification and post-level accuracy is around 90%. While there is room for improvement, this variant of the system is accurate enough to enable analysis of the entire Darkode forum with automatic annotation.

Throughout the rest of this work, we focus on NP-level evaluation and post-level NP accuracy.

## 5 Domain Adaptation

Table 2 only showed results for training and evaluating within the same forum (Darkode). However, we wish to apply our system to extract product occurrences from a wide variety of forums, so we are interested in how well the system will generalize to a new forum. Tables 3 and 4 show full results of several systems in within-forum and cross-forum evaluation settings. Performance is severely degraded in the cross-forum setting compared to the within-forum setting, e.g., on NP-level $F_1$, a Hack Forums-trained model is 14.4 $F_1$ worse at the Darkode task than a Darkode-trained model (63.2 vs. 77.6). Differences in how the systems adapt between different forums will be explored more thoroughly in Section 5.4.

In the next few sections, we explore several possible methods for improving results in the cross-forum settings and attempting to build a more domain-general system. These techniques generally reflect two possible hypotheses for the difficult cross-domain effects:

**Hypothesis 1:** Product inventories are the primary difference across domains; context-based features will transfer, but the main challenge is not being able to recognize unknown products.

| | Darkode | Hack Forums | Blackhat | Nulled |
| | Trained on Darkode | | | |
| Dict | 62.5 | 43.3 | 48.3 | 61.6 |
| Post | 95.8 | 67.6 | 75.8 | 82.5 |
| +Brown | 92.7 | 69.8 | 75.8 | 86.0 |
| +Gaz | 93.7 | 71.3 | – | – |
| | Trained on Hack Forums | | | |
| Dict | 51.0 | 52.2 | 51.6 | 56.9 |
| Post | 84.3 | 80.8 | 80.6 | 83.7 |
| +Brown | †89.5 | 81.6 | 80.6 | 83.7 |
| +Gaz | 85.4 | †85.2 | – | – |

Table 4: Test set results at the whole-post level in within-forum and cross-forum settings for a variety of different systems. Here, Brown clusters and gazetteers may be slightly more impactful; daggers indicate statistically significant gains over the post-level system with $p < 0.05$ according to a bootstrap resampling test.

**Hypothesis 2:** Product inventories and stylistic conventions both differ across domains; we need to capture both to adapt models successfully.

### 5.1 Brown Clusters

To test Hypothesis 1, we investigate whether additional lexical information helps identify product-like words in new domains. A classic semi-supervised technique for exploiting unlabeled target data is to fire features over word clusters or word vectors (Turian et al., 2010). These features should generalize well across domains that the clusters are formed on: if product nouns occur in similar contexts across domains and therefore wind up in the same cluster, then a model trained on domain-limited data should be able to learn that that cluster identity is indicative of products.

We form Brown clusters on our unlabeled data from both Darkode and Hack Forums (see Table 1

for sizes). We use Liang (2005)'s implementation to learn 50 clusters.[8] Upon inspection, these clusters do indeed capture some of the semantics relevant to the problem: for example, the cluster 110 has as its most frequent members *service*, *account*, *price*, *time*, *crypter*, and *server*, many of which are product-associated nouns. We incorporate these as features into our model by characterizing each token with prefixes of the Brown cluster ID; we used prefixes of length 2, 4, and 6.

Tables 3 and 4 show the results of incorporating Brown cluster features into our trained models. Overall, these features lead almost uniformly to slight improvements on NP-level product $F_1$. Results are more mixed on post-level accuracy, and overall, only one result improves significantly. This indicates that Brown clusters might be a useful feature, but do not solve the domain adaptation problem in this context.[9]

## 5.2 Type-level Annotation

Another approach following Hypothesis 1 is to use small amounts of supervised data, One cheap approach for annotating data in a new domain is to exploit type-level annotation (Garrette and Baldridge, 2013; Garrette et al., 2013). Our token-level annotation standard is relatively complex and time-consuming, but a researcher could quite easily provide a few exemplar products for a new forum based on just a few minutes of reading posts and analyzing the forum.

Given the data that we've already annotated, we can simulate this process by iterating through our labeled data and collecting annotated product names that are sufficiently common. Specifically, we take all (lowercased, stemmed) product tokens and keep those occurring at least 4 times in the training dataset (recall that these datasets are $\approx 700$ posts). This gives us a list of 121 products in Darkode and 105 products in Hack Forums.

To incorporate this information into our system, we add a new feature on each token indicating whether or not it occurs in the gazetteer. At training time, we scrape the gazetteer from the training set. At test time, we use the gazetteer from

---

[8]The number of clusters was selected based on experimentation with our various development sets.

[9]Vector representations of words are also possible here; however, our initial experiments with these did not show better gains than with using Brown clusters. That is consistent with the results of Turian et al. (2010) who showed similar performance between Brown clusters and word vectors for chunking and NER.

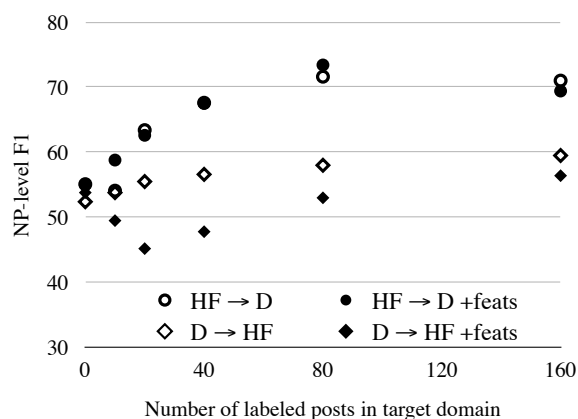

Figure 2: Token-supervised domain adaptation results for two settings. As our system is trained on an increasing amount of target-domain data (x-axis), its performance generally improves. However, adaptation from Hack Forums to Darkode is much more effective than the other way around, and using domain features as in Daume III (2007) gives little benefit over naïve use of the new data.

the target test domain as a form of partial type-level supervision. Tables 3 and 4 shows the results of incorporating the gazetteer into the system. In aggregate, gazetteers appear to provide a slight gain over the baseline system, like Brown clusters, though many of these individual improvements are not statistically significant.

## 5.3 Token-level Annotation

We now turn our attention to methods that might address Hypothesis 2. If we assume the domain transfer problem is more complex, we really want to leverage labeled data in the target domain rather than attempting to transfer features based only on type-level information. Specifically, we are interested in cases where a relatively small number of labeled posts (less than 100) might provide substantial benefit to the adaptation; a researcher could plausibly do this annotation in a few hours.

We consider two ways of exploiting labeled target-domain data. The first is to simply take these posts as additional training data. The second is to also employ the "frustratingly easy" domain adaptation method of Daume III (2007). In this framework, each feature fired in our model is actually fired twice: one copy is domain-general and one is conjoined with the domain label (here, the name of the forum).[10] In doing

---

[10]If we are training on data from $k$ domains, this gives rise

| | Test | Darkode | | | Hack Forums | | | Blackhat | | | Nulled | | |
|---|---|---|---|---|---|---|---|---|---|---|---|---|---|
| System | | % OOV | $R_{seen}$ | $R_{oov}$ | % OOV | $R_{seen}$ | $R_{oov}$ | % OOV | $R_{seen}$ | $R_{oov}$ | % OOV | $R_{seen}$ | $R_{oov}$ |
| Binary (Darkode) | | 21 | 82 | 67 | 41 | 69 | 51 | 44 | 73 | 51 | 33 | 73 | 50 |
| Binary (HF) | | 52 | 75 | 35 | 36 | 77 | 40 | 54 | 71 | 36 | 38 | 79 | 37 |

Table 5: Product token OOV rates on development sets (test set for Blackhat and Nulled) of various forums with respect to training on Darkode and Hack Forums. We also show the recall of an NP-level system on seen ($R_{seen}$) and OOV ($R_{OOV}$) tokens. Darkode seems to be more "general" than Hack Forums: the Darkode system generally has lower OOV rates and provides more consistent performance on OOV tokens than the Hack Forums system.

so, the model should gain some ability to separate domain-general from domain-specific feature values. For both training methods, we upweight the contribution of the target-domain posts in the objective by a factor of 5.

Figure 2 shows learning curves for both of these methods in two adaptation settings as we vary the amount of labeled target-domain data. The system trained on Hack Forums is able to make good use of labeled data from Darkode: having access to 20 labeled posts leads to gains of roughly 7 $F_1$. Interestingly, the system trained on Darkode is not able to make good use of labeled data from Hack Forums, and the domain-specific features actually cause a substantial drop in performance until we have included a substantial amount of data from Hack Forums. This likely indicates we are over-fitting the small Hack Forums training set with the domain-specific features.

### 5.4 Analysis

In order to understand the variable performance and shortcomings of the domain adaptation approaches we explored, it is useful to examine our two initial hypotheses and characterize the datasets a bit further. To do so, we break down system performance on products seen in the training set versus novel products. Because our systems depend on lexical and character $n$-gram features, we expect that they will do better at predicting products we have seen before.

Table 5 confirms this intuition: it shows product out-of-vocabulary rates in each of the four forums relative to training on both Darkode and Hack Forums, along with recall of a NP-level system on both previously seen and OOV products. As expected, performance is substantially higher on in-vocabulary products. Interestingly, OOV rates of a Darkode-trained system are generally lower, indicating that that forum has better all-around product

to up to $k + 1$ total versions of each feature.

coverage. A system trained on Darkode is therefore in some sense more domain-general than one trained on Hack Forums.

This would seem to confirm Hypothesis 1. Table 3 shows that the Hack Forums-trained system achieves a 20% error reduction on Hack Forums compared to a Darkode-trained system. In contrast, the Darkode-trained system obtains a 43% error reduction on Darkode relative to a Hack Forums-trained system. Darkode's better product coverage helps explain why Section 5.3 showed better performance of adapting Hack Forums to Darkode than vice versa: augmenting Hack Forums data with a few posts from Darkode can give critical knowledge about new products, but this is less true if the forums are reversed. Duplicating features and adding parameters to the learner also has less of a clear benefit when adapting from Darkode, when the types of knowledge that need to be added are less concrete.

Note, however, that these results do not tell the full story. Table 5 reports recall values, but not all systems have the same precision/recall trade-off: although they were tuned to balance precision and recall on their respective development sets, the Hack Forums-trained system is much more precision-oriented on Nulled than the Darkode-trained system. In fact, Table 3 shows that the Hack Forums-trained system actually performs better on Nulled. This indicates that there is some truth to Hypothesis 2: product coverage is not the only important factor determining performance.

## 6 Conclusion

We present a new dataset of posts from cybercrime marketplaces annotated with product references, a task which blends IE and NER. Learning-based methods degrade in performance when applied to new forums, and while we explore methods for fine-grained domain adaption in this data, effective methods for this task are still an open question.

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
