# Peer review of "Identifying Products in Online Cybercrime Marketplaces: A Dataset and Fine-grained Domain Adaptation Task"

_ACL 2017 — decision unknown_

[Official Review · Reviewer 1 · rating 3 · confidence 4]
soundness 3 · originality 3 · clarity 5 · impact 3 · substance 5 · appropriateness 5 · meaningful comparison 3 · presentation format Poster

This paper presents a new dataset with annotations of products coming from
online cybercrime forums. The paper is clear and well-written and the
experiments are good. Every hypothesis is tested and compared to each other.

However, I do have some concerns about the paper:

1. The authors took the liberty to change the font size and the line spacing of
the abstract, enabling them to have a longer abstract and to fit the content
into the 8 pages requirement.

2. I don't think this paper fits the tagging, chunking, parsing area, as it is
more an information extraction problem.

3. I have difficulties to see why some annotations such as sombody in Fig. 1
are related to a product.

4. The basic results are very basic indeed and - with all the tools available
nowadays in NLP -, I am sure that it would have been possible to have more
elaborate baselines without too much extra work.

5. Domain adaptation experiments corroborate what we already know about
user-generated data where two forums on video games, e.g., may have different
types of users (age, gender, etc.) leading to very different texts. So this
does not give new highlights on this specific problem.